# Association of Serum Uric Acid Levels with Cardiometabolic Factors in Adolescents with Obesity: A Cross-Sectional Study

**DOI:** 10.3390/metabo15040237

**Published:** 2025-03-31

**Authors:** Miguel Angel Villasis-Keever, Jessie Nallely Zurita-Cruz, Iris Alejandra Alcaraz-Hurtado, Miguel Klünder-Klünder, Jenny Vilchis-Gil, Ana Laura Romero-Guerra, Ana Laura López-Beltran, Martha Alicia Delgadillo-Ruano

**Affiliations:** 1Analysis and Synthesis of the Evidence Research Unit, National Medical Center XXI Century, Instituto Mexicano del Seguro Social, Mexico City 06720, Mexico; miguel.villasis@gmail.com; 2Facultad de Medicina Universidad Nacional Autónoma de Mexico, Hospital Infantil de Mexico Federico Gómez, Mexico City 06720, Mexico; 3Department of Pediatric, National Medical Center XXI Century, Instituto Mexicano del Seguro Social, Mexico City 06720, Mexico; iris_1604@hotmail.com (I.A.A.-H.); laura1141@gmail.com (A.L.R.-G.); 4Epidemiological Research Unit in Endocrinology and Nutrition, Hospital Infantil de México Federico Gómez, Ministry of Health (SSA), Mexico City 06720, Mexico; klunderk@gmail.co (M.K.-K.); jvilchisgil@gmail.com (J.V.-G.); 5Department of Pediatric Endocrinology, UMAE Pediatric Hospital of the National Medical Center West, IMSS, Guadalajara 44360, Mexico; analau78@hotmail.com (A.L.L.-B.); adelgadilloruano@yahoo.com (M.A.D.-R.)

**Keywords:** obesity, uric acid, insulin resistance, cardiometabolic factors, adolescent

## Abstract

Introduction: High serum uric acid (SUA) levels are known to be correlated with cardiometabolic factors in adults, but this relationship is less clear in the pediatric population, particularly given the undefined cutoff points for high SUA levels. Objetive: This study aimed to explore the associations between SUA levels and cardiometabolic factors in obese adolescents. Materials and Methods: We conducted a cross-sectional study of 391 adolescents aged 10 to 18 years with obesity (BMI > 95th percentile), assessing outcomes such as hypertriglyceridemia, reduced HDL cholesterol, hypertension, hyperglycemia, and insulin resistance. The statistical methods used to compare SUA levels with cardiometabolic factors included the Mann-Whitney U test and the chi-square test. Results: The results revealed that the median SUA level was 5.9 mg/dL, with significant differences between the sexes (5.5 mg/dL for girls and 6.1 mg/dL for boys). The highest SUA tertile (≥6.41 mg/dL) was associated with a significantly greater frequency of hyperglycemia and hypertriglyceridemia than the lowest tertile (*p* < 0.005). Conclusions: The study concluded that higher SUA levels are significantly associated with specific cardiometabolic risks in adolescents with obesity, highlighting the importance of monitoring SUA levels in this population.

## 1. Introduction

Childhood obesity has become a significant public health issue worldwide. As highlighted by the World Health Organization (WHO), obesity is currently a prominent challenge because it plays a pivotal role as a primary risk factor for cardiometabolic events [1]. Moreover, insulin resistance (IR) and a proinflammatory environment have been identified as key components in the pathophysiology of obesity [2]. The prevalence of overweight status and obesity in Mexico has consistently increased in recent years [3]. These conditions have been strongly linked to cardiovascular and metabolic diseases [4].

Uric acid is the final metabolic byproduct derived from both endogenous and exogenous purines. The production of uric acid involves the oxidation of its immediate precursor, xanthine, with serum levels regulated by a delicate balance between synthesis and excretion [5,6]. In pediatric populations, serum uric acid (SUA) levels fluctuate with age and typically peak during early adolescence. Sex also influences SUA levels, although a consensus for defining normal ranges has not been reached, with variations across different studies [7,8]. This sex difference can be explained by the fact that estrogens increase the renal clearance of urate and reduce the post-secretory tubular reabsorption of urate [9]. Some studies have proposed thresholds for hyperuricemia, setting it at levels exceeding 6.0 mg/dL in women and 7.7 mg/dL in men [9].

Uric acid induces oxidative stress and disrupts the expression of insulin-like growth factor-I (IGF-I), leading to diminished hepatic IGF levels. Consequently, this impairs renal plasma flow and the glomerular filtration of uric acid, thus interfering with insulin metabolism and fostering insulin resistance [10,11].

Moreover, elevated SUA levels have been correlated with systemic arterial hypertension (SAH) [12]. The pathophysiology of SAH involves decreased renal blood flow, increased lactate pathway activity due to ischemia from irreversible damage to small renal vessels, and activation of the renin-angiotensin system, which disrupts uric acid elimination [12,13]. Investigations in pediatric populations have revealed a positive association between SUA levels and serum lipid concentrations, suggesting that increased SUA levels disrupt adipogenesis and are linked to obesity and altered anthropometric indices [7,14].

Although the associations between elevated uric acid levels and dyslipidemia, hyperglycemia, and SAH, conditions which are driven by oxidative stress, have been observed in obese adolescents, the SUA level has not yet been used as a metabolic syndrome (MetS) screening criterion [7,8,15,16]. One rationale for this omission is the variability of SUA levels with sex and pubertal stage.

Hence, the present study aims to elucidate the relationships between the SUA concentration, insulin resistance, and MetS in adolescents with obesity.

## 2. Materials and Methods

### 2.1. Study Design

This cross-sectional study was conducted between January 2019 and May 2022 at four tertiary care pediatric centers in Mexico (Hospital Infantil de Mexico Federico Gómez, Pediatric Hospital Centro Médico Nacional Siglo XXI, Pediatric Hospital of the National Medical Center West Guadalajara and High Specialty South Central Hospital of Petroleos Mexicanos). Participants aged 10–18 years who were diagnosed with obesity (BMI > 95th percentile based on the 2000 CDC growth charts) [17] were enrolled. The exclusion criteria included the presence of any concomitant conditions, elevated serum creatinine levels, a general urine test showing proteinuria, or any other abnormalities, the use of medications known to potentially influence weight or appetite (such as genetic syndromes, steroids, fluoxetine, insulin sensitizers, insulin, anorexigenics, or intestinal fat absorption inhibitors), and refusal to participate. Data were collected on several variables, including anthropometric measurements and fasting plasma concentrations of high-density lipoprotein cholesterol (HDLc), low-density lipoprotein cholesterol (LDLc), triglycerides (TGLs), glucose, insulin, and SUA.

Sexual maturity was assessed by a pediatric endocrinologist with the Tanner scale.

### 2.2. Anthropometry

A certified nutritionist recorded the anthropometric indicators of each subject. We conducted an anthropometric assessment by measuring key parameters, including height (cm), weight (kg), and body mass index (BMI). BMI is a calculation used to evaluate body shape and is derived from an individual’s weight and height using the formula: BMI = weight (kg)/[height (m)]^2^. Height was measured with a stadiometer (SECA Model 769). Weight was measured with the bioimpedance method (Tanita BC-568 Segmental Body Composition Monitor, Tokyo, Japan) with the patient fasting between 7:00 and 8:00 a.m., barefoot, and wearing only underwear.

### 2.3. Cardiometabolic Profile Assessment and Uric Acid Level Measurements

After a minimum of 12 h of fasting, between 7:00 and 8:00 a.m., blood samples were collected from the antecubital vein in the forearm following the anthropometric assessment. Glucose, TGL, HDLc, and uric acid levels were determined by colorimetric enzymatic methods (Bayer Diagnostics, Puteaux, France). Insulin was measured by chemiluminescence (Roche-Hitachi Modular P and D). Intra- and interassay coefficients of variation <7% were considered acceptable. A standard curve was also generated for each assay.

### 2.4. Definitions

The insulin resistance index (HOMA-IR) was calculated according to the following formula: HOMA-IR = fasting glucose (mg/dL) × fasting insulin (µU/mL)/405. The HOMA-IR cutoff point for the diagnosis of IR was 3.1 [18]. Hypertension was defined as either systolic or diastolic blood pressure exceeding the 95th percentile for the individual’s age and sex. Hyperglycemia was defined as a glucose level ≥100 mg/dL. Hypertriglyceridemia was defined as a triglyceride level ≥150 mg/dL [19,20]. In addition, HDLc levels were considered reduced as follows: HDLc < 40 mg/dL in males and <50 mg/dL in females, as recommended by the International Diabetes Federation (IDF) [19,20]. Abdominal obesity was defined as a waist circumference exceeding the 95th percentile for the individual’s age and sex. MetS was defined as having abdominal obesity and two or more of the following cardiometabolic abnormalities, according to the definitions already mentioned above: hypertension, dyslipidemia (either hypertriglyceridemia or reduced HDLc), or elevated fasting plasma glucose [19,20]. Pubertal development was classified as follows: Tanner stage 1, prepubertal; and Tanner stages 2–5, pubertal.

### 2.5. Statistical Analysis

The Kolmogorov-Smirnov test indicated that the distribution of continuous variables was nonparametric. These variables are represented by the median and interquartile range (IQR), and differences between groups were analyzed with the Mann-Whitney U test. Categorical variables are presented as frequencies and percentages, and differences between groups were analyzed with the chi-square test.

Owing to the lack of international consensus on the specific cut-off for hyperuricemia in pediatric patients and the influence of age and sex on serum levels, patients were categorized on the basis of SUA tertiles (T1, T2, and T3). To compare quantitative variables across these tertiles, the Kruskal-Wallis test was employed.

The patients were subsequently segmented into two groups: those with SUA levels in T1 or T2 (<6.40 mg/dL) and those with SUA levels in T3 (>6.41 mg/dL). A sex-stratified analysis set the T3 cutoffs at ≥6.11 mg/dl for women and ≥6.54 mg/dL for men.

The odds ratios (ORs) for cardiometabolic factors and MetS in individuals within SUA T3 were calculated using the Cochran–Mantel–Haenszel test with 95% confidence intervals (CIs).

The cutoff values for SUA were determined with receiver operating characteristic (ROC) curves for each cardiometabolic factor, including hyperglycemia, reduced HDLc, hypertriglyceridemia, hypertension, IR, and MetS. The sensitivity, specificity, correct classification, and area under the curve (AUC) were calculated for each parameter.

Statistical significance was set to *p*-values < 0.05. All the analyses were conducted using STATA version 12.0.

### 2.6. Ethical Considerations

The study protocol complied with the principles of the Declaration of Helsinki and was approved by the National Research and Health Ethics Committee of the Mexican Social Security Institute (IMSS) (registry number R-2019-3603-050). The parents/caregivers provided written informed consent, and each child provided assent.

## 3. Results

### 3.1. General Characteristics of the Adolescents

Among the 531 adolescent candidates who were initially identified, nine were excluded because of incomplete laboratory data, 112 were excluded because they were overweight, and 28 were excluded because they declined to participate (Figure 1). Ultimately, 391 adolescents with obesity (47.6% female) were enrolled. The median age of the participants was 12 years, with a median BMI z score of 2.2. On the basis of the puberty stage assessment, 91.6% of the subjects (*n* = 358) were classified as pubertal (Tanner stages 2–5) (Table 1).

In our cohort, the median serum TGL level exceeded 150 mg/dL, whereas the median HDLc level was less than 40 mg/dL (Table 1). The median SUA level was 5.9 mg/dL overall, with a median of 6.1 mg/dL in males and 5.5 mg/dL in females. Among the patients, 79.2% presented with altered HDLc levels (*n* = 310), 44.5% presented with hypertriglyceridemia (*n* = 174), 20.2% presented with abnormal serum glucose values, and 39.1% met the criteria for MetS.

The analysis of the biochemical parameters by sex revealed that boys had significantly higher serum glucose levels than girls did (92.0 mg/dL vs. 88.5 mg/dL, *p* < 0.001), corresponding with a higher incidence of hyperglycemia among boys than girls (24.4% vs. 15.6%, *p* = 0.030). Although HDLc levels were similar between boys and girls (37.0 mg/dl for both), differing criteria for reduced HDLc according to sex meant that a greater proportion of girls than boys had this form of dyslipidemia (91.4% vs. 68.3%, *p* > 0.001).

### 3.2. Comparison Between Groups

In the entire study population, the median SUA level was 5.9 mg/dL (IQR 4.8–6.8 mg/dL). Among girls, the median SUA was 5.5 mg/dL (IQR 4.6–6.5 mg/dL), and among boys, it was 6.1 mg/dL (IQR 5.0–7.0 mg/dL). Comparisons of biochemical profiles and cardiometabolic alterations based on SUA tertiles revealed that those in the highest tertile of SUA had increased serum glucose and TGL levels, along with a higher prevalence of hyperglycemia, hypertriglyceridemia, and MetS (Table 2).

A comparison of subjects across SUA tertiles revealed that patients in T3 were significantly older and had higher BMI z scores than those in T1 and T2. Additionally, individuals in T3 presented with higher serum triglyceride (160.0 mg/dL vs. 135.0 mg/dL, *p* = 0.002) and glucose levels (93.0 mg/dL vs. 89.0 mg/dL, *p* < 0.001) than those in T1 and T2.

Stratified by sex, female adolescents with SUA > 6.11 mg/dL presented with higher glucose levels and a higher incidence of MetS, whereas males with SUA > 6.54 mg/dL presented with a greater frequency of hyperglycemia, hypertriglyceridemia, and MetS (Table 3).

With respect to SUA T3, the ORs for cardiometabolic factors and MetS were calculated and found to be significantly elevated for abdominal obesity (OR 2.26), hyperglycemia (OR 2.99), hypertriglyceridemia (OR 2.19), and MetS (OR 2.35) (Figure 2). Sex-specific analysis revealed differences in the impact of MetS risk factors between boys and girls, with a notably greater increase in the OR for MetS associated with SUA levels in boys (OR 2.34, 95% CI 1.29–4.42, *p* = 0.004) than in girls (OR 2.20, 95% CI 1.17–4.12, *p* = 0.012).

Finally, an exploratory analysis was conducted with ROC curves, which identified a SUA cutoff point of >6.8 mg/dL for detecting individual cardiometabolic factors. This cutoff demonstrated high specificity (>74%) but low sensitivity (<42%). Specifically, for the detection of MetS, the specificity was 82.2%; for hyperglycemia, it was 78.1%; and for hypertriglyceridemia, it was 82.5%. In contrast, lower AUC values were observed for other factors, such as insulin resistance, altered HDL-C levels, and hypertension (Table 4).

## 4. Discussion

In recent years, elevated SUA levels have emerged as independent predictors of DM2, IR, fatty liver, dyslipidemia, and MetS [21]. This phenomenon appears to be influenced by the development of mitochondrial oxidative stress and impaired insulin-mediated stimulation of nitric oxide production by endothelial cells [21,22].

An SUA level greater than 7 mg/dL is commonly considered high in adults. With respect to cardiometabolic factors, research has focused primarily on the association between obesity and MetS but typically has not addressed specific uric acid levels, including studies within the Mexican population. To establish detailed cutoff points related to uric acid and MetS in adults, further research specifically targeting these parameters is needed [23,24,25,26,27].

A clear reference value for SUA levels in the pediatric population is lacking because these levels gradually increase from birth to the end of school age [6,9]. Studies in adolescents have suggested that SUA values exceeding 5.2 mg/dL are more frequently associated with hypertension, hyperinsulinemia, obesity, and reduced HDL cholesterol [28,29]. Other investigations have proposed sex-specific thresholds, identifying levels above the 95th percentile (6.8 mg/dL for men and 5.4 mg/dL for women) as indicative of increased cardiovascular risk [8,30]. Moreover, for every 1 mg/dL increase in SUA, there is a 54% increased risk of MetS [9]. Notably, studies by Safiri et al. and Seo et al. reported associations between elevated SUA levels and a higher frequency of MetS in adolescents [7,8].

Obesity has been associated with an inflammatory state leading to elevated SUA levels, oxidative stress, and an increased risk of metabolic alterations, although the chronological relationship remains controversial [9,29,31]. Our study revealed higher SUA levels in males than in females, which is consistent with previous findings attributed to increased body mass and testosterone levels in men, alongside decreased urate excretion during puberty [32,33].

The biological mechanisms underlying the association between SUA and MetS remain unclear. Some researchers have suggested that high intake of purine- and fructose-rich foods contributes to elevated SUA levels, obesity, and MetS development [9,34]. Adiponectin, an anti-inflammatory and antiatherogenic peptide synthesized in adipose tissue, may also play a role, with decreased levels associated with increased SUA levels, oxidative stress, and inflammation, contributing to cardiovascular and metabolic diseases such as diabetes mellitus and MetS [35,36].

With respect to the more adverse cardiometabolic profile identified in boys compared to girls, it is crucial to note that more than 90% of the subjects were adolescents, and the effects of hormones significantly influence these differences. Estrogens lead to a different distribution of body fat. In individuals with prediabetes or diabetes, men exhibit a less favorable cardiometabolic profile, which is associated with lower lean mass and greater body fat than women [37]. The interconnection of these factors manifests as follows: uric acid triggers IR in pancreatic beta cells via the IRS2/AKT pathway and promotes fat accumulation in liver cells. This accumulation disrupts insulin signaling through the activation of the NOD-like receptor family pyrin domain containing 3 inflammasome. Consequently, the quantity of abdominal fat is a crucial marker for predicting SUA levels in individuals newly diagnosed with DM2 [38,39].

Given the high specificity achieved with a serum uric acid (SUA) cutoff of >6.8 mg/dL for detecting metabolic syndrome, hyperglycemia, and hypertriglyceridemia, this parameter could prove useful in monitoring adolescents to assess changes in these cardiometabolic conditions without requiring a full lipid profile. This approach is particularly beneficial in primary care settings where testing for SUA is more feasible than conducting comprehensive lipid profiling. This review aims to elucidate the utility of SUA as a simpler alternative to full lipid profiles and is especially suitable for resource-constrained or primary care environments.

Our study’s limitations include its cross-sectional design, which prevents the longitudinal observation of changes in SUA levels, the fact that glycated hemoglobin testing was not included for the diagnosis of hyperglycemia, and the absence of detailed information on dietary and physical activity habits. Another limitation is that by including only obese patients, the SUA cutoff point is specific but lacks sensitivity for detecting cardiometabolic factors, as observed in the results. Therefore, this cutoff can be applied only to adolescents with obesity. While we do not have body composition data to substantiate our hypothesis about the impact of hormones on the cardiometabolic profile through differences in body composition between sexes, evidence of a positive correlation between SUA levels and visceral fat has been established [40]. Nonetheless, the strength of the study lies in the homogeneity of the sample population regarding sociodemographic characteristics, enhancing the internal validity of our findings.

## 5. Conclusions

SUA levels in the 3rd tertile (>6.41 mg/dL, >6.11 mg/dL in women, and >6.54 mg/dL in men) are associated with hyperglycemia and MetS in adolescents with obesity. These findings suggest that measuring SUA levels could aid in the early identification of insulin resistance and MetS in this population.

## Figures and Tables

**Figure 1 metabolites-15-00237-f001:**
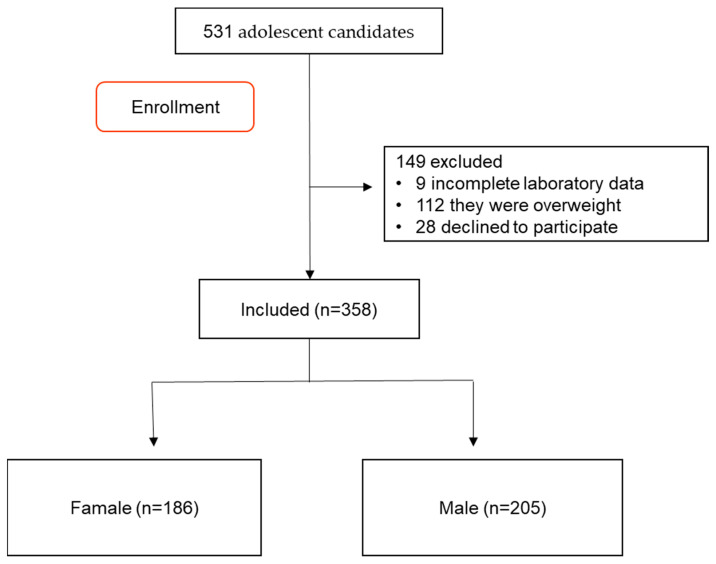
Flow diagram.

**Figure 2 metabolites-15-00237-f002:**
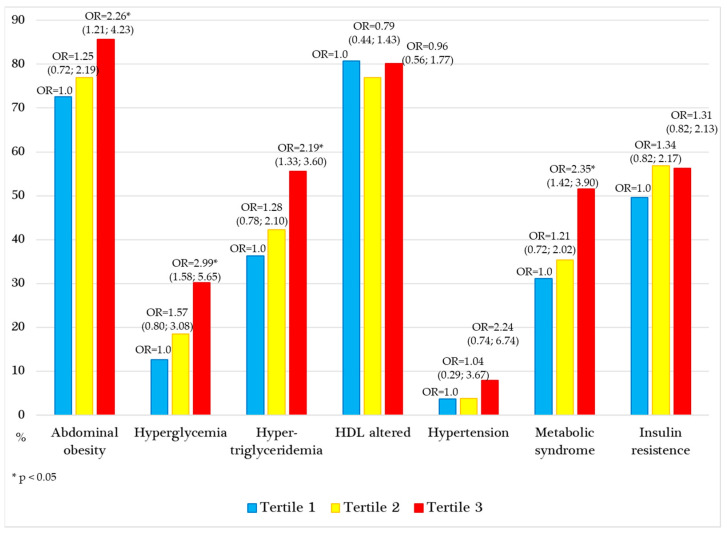
Proportion and odds ratio (OR) of cardiometabolic alterations and metabolic syndrome based on serum uric acid tertiles.

**Table 1 metabolites-15-00237-t001:** General characteristics and lipid profiles of subjects.

	All	Female	Male	
	(*n* = 391)	(*n* = 186)	(*n* = 205)	*p*
	Median (Interquartile Range)			
Age (years)	12.0 (10.0, 15.0)	13.5 (11.0, 15.0)	12.0 (10.0, 14.0)	0.001
Female sex *	186 (47.6)			
Score Z del IMC	2.2 (1.9, 2.4)	2.0 (1.8, 2.3)	2.2 (2.0, 2.4)	0.001
Waist circumference (cm)	94.9 (87.0, 102.0)	92.7 (86.0, 101.0)	97.0 (88.0, 102.0)	0.036
Puberty *				0.006
Prepubertal	33 (8.4)	10 (5.4)	23 (11.2)	
Pubertal	358 (91.6)	176 (94.6)	182 (88.8)	
Uric acid (mg/dL)	5.9 (4.8, 6.9)	5.5 (4.6, 6.5)	6.1 (5.0, 7.0)	<0.001
Glucose (mg/dL)	91.0 (85.0, 98.0)	88.5 (84.0, 96.0)	92.0 (87.0, 99.0)	<0.001
Total Cholesterol (mg/dL)	160.0 (138.0, 179.0)	159.0 (141.0, 180.0)	160.0 (138.0, 176.0)	0.771
Triglycerides (mg/dL)	141.0 (98.0, 175.0)	141.5 (100.0, 174.0)	140.0 (98.0, 175.0)	0.956
HDL Cholesterol (mg/dL)	37.0 (32.0, 42.0)	37.0 (33.0, 42.0)	37.0 (32.0, 42.0)	0.710
LDL Cholesterol (mg/dL)	93.2 (75.0, 109.0)	93.0 (76.0, 110.0)	93.6 (73.6, 108.0)	0.659
Insulin (µu/mL)	14.9 (8.2, 23.8)	15.4 (9.9, 23.8)	14.9 (6.9, 23.8)	0.099
HOMA-IR	3.4 (1.9, 5.8)	3.4 (2.1, 5.8)	3.3 (1.6, 5.8)	0.150
Abdominal obesity	306 (78.3)	138 (74.2)	168 (81.9)	0.063
Hyperglycemia *	79 (20.2)	29 (15.6)	50 (24.4)	0.030
Hypertriglyceridemia *	174 (44.5)	83 (44.6)	91 (44.4)	0.963
Altered HDL *	310 (79.2)	170 (91.4)	140 (68.3)	<0.001
Insulin resistance *	212 (54.2)	107 (57.5)	105 (51.2)	0.211
Hypertension *	20 (5.1)	7 (3.8)	13 (6.3)	0.248
Metabolic syndrome *	153 (39.1)	75 (40.3)	78 (38.0)	0.645

* Frequency (%).

**Table 2 metabolites-15-00237-t002:** Comparison of the lipid profiles of subjects in the 1st to 3rd serum uric acid tertiles.

	Serum Uric Acid Tertile			*p*
	T1 <5.20 mg/dL (*n* = 135)	T2 5.21–6.40 mg/dL (*n* = 130)	T3 >6.41 mg/dL (*n* = 126)	
	Median (Interquartile Range)			
Female sex *	79 (58.5)	59 (45.4)	48 (38.1)	0.004
Male sex	56 (41.5)	71 (54.6)	78 (61.9)	
Score Z del IMC	2.0 (1.8, 2.4)	2.2 (2.0, 2.4)	2.2 (2.0, 2.4)	0.026
Waist circumference (cm)	91.2 (84.0, 100.0)	93.2 (87.0, 101.0)	98.0 (91.5, 105.0)	<0.001
Glucose (mg/dL)	87.0 (83.0, 94.0)	91.2 (86.0, 98.0)	93.0 (88.0, 101.0)	<0.001
Total Cholesterol (mg/dL)	157.0 (138.0, 173.0)	160.0 (138.0, 180.0)	159.5 (140.0, 180.0)	0.453
Triglycerides (mg/dL)	134.0 (91.0, 165.0)	137.0 (98.0, 170.0)	160.0 (108.0, 194.0)	0.006
HDL Cholesterol (mg/dL)	37.8 (34.0, 43.0)	36.5 (32.0, 41.0)	37.0 (32.0, 42.0)	0.129
LDL Cholesterol (mg/dL)	93.6 (74.8, 107.0)	94.0 (74.2, 110.0)	93.0 (75.6, 110.0)	0.918
Insulin (µu/mL)	14.3 (9.1, 21.2)	14.9 (8.4, 23.8)	16.1 (7.4, 28.5)	0.366
HOMA-IR	3.0 (2.0, 4.7)	3.5 (2.9, 6.0)	3.6 (1.8, 6.6)	0.143
Abdominal obesity	98 (72.6)	100 (76.9)	108 (85.7)	0 033
Hyperglycemia	17 (12.6)	24 (18.5)	38 (30.2)	0.002
Hypertriglyceridemia	49 (36.3)	55 (42.3)	70 (55.6)	0.006
Altered HDL *	109 (80.7)	100 (76.9)	101 (80.2)	0.714
Insulin resistance	67 (49.6)	74 (56.9)	71 (56.3)	0.415
Hypertension	5 (3.7)	5 (3.8)	10 (7.9)	0.217
Metabolic syndrome	42 (31.1)	46 (35.4)	65 (51.6)	0.002

* Frequency (%).

**Table 3 metabolites-15-00237-t003:** Comparison of the general characteristics and lipid profiles of subjects according to sex and serum uric acid tertile.

	Female (*n* = 186)			Male (*n* = 205)		
	Serum Uric Acid Tertile			Serum Uric Acid Tertile		
	T1–T2 <6.10 mg/dL (*n* = 126)	T3 ≥6.11 mg/dL (*n* = 60)	*p*	T1–T2 <6.53 mg/dL (*n* = 135)	T3 ≥6.54 mg/dL (*n* = 70)	*p*
	Median (Interquartile Range)			Median (Interquartile Range)		
Score Z del IMC	2.0 (1.8, 2.3)	2.1 (2.0, 2.4)	0.002	2.2 (2.0, 2.5)	2.2 (2.0, 2.4)	0.608
Waist circumference (cm)	89.8 (84.8, 98.0)	98 (91.6, 105.0)	<0.001	94.5 (86.1, 101.0)	99.0 (93.0, 106.0)	<0.001
Glucose (mg/dL)	87.0 (83.0, 95.0)	92.0 (87.0, 100.0)	0.002	91.0 (86.0, 98.0)	94.0 (89.0, 102.0)	0.071
Total Cholesterol (mg/dl)	159.5 (140.0, 180.0)	158.0 (145.0, 179.1)	0.895	158.0 (138.0, 172.0)	165.0 (140.0, 184.0)	0.101
Triglycerides (mg/dL)	136.5 (92.0, 174.0)	156.0 (111.0, 174.5)	0.234	129.0 (91.0, 164.0)	166.0 (111.0, 231.0)	<0.001
HDL Cholesterol (mg/dL)	37.0 (33.0, 42.0)	36.5 (32.0, 41.5)	0.440	37.0 (33.0, 42.0)	37.0 (30.0, 43.0)	0.682
LDL Cholesterol (mg/dL)	94.0 (75.0, 111.0)	93.0 (81.5, 106.2)	0.774	93.4 (73.0, 105.1)	95.0 (77.0, 110.0)	0.260
Insulin (µu/mL)	16.3 (11.0, 23.0)	15.0 (6.5, 27.9)	0.426	13.5 (6.6, 21.2)	18.2 (7.4, 29.2)	0.117
HOMA-IR	3.5 (2.3, 5.1)	3.3 (1.5, 6.3)	0.552	3.0 (1.5, 4.8)	4.4 (1.8, 7.1)	0.072
Abdominal obesity *	84 (66.7)	54 (90.0)	0.001	110 (81.5)	58 (82.9)	0.808
Hyperglycemia *	13 (10.3)	16 (26.7)	0.004	27 (20.0)	23 (32.9)	0.042
Hypertriglyceridemia *	52 (41.3)	31 (51.7)	0.182	49 (36.3)	42 (60.0)	<0.001
Altered HDL*	115 (91.3)	55 (91.7)	0.928	90 (66.7)	50 (71.4)	0.487
Insulin resistance *	73 (57.9)	34 (56.7)	0.870	66 (48.9)	39 (55.7)	0.354
Hypertension *	5 (4.0)	2 (3.3)	0.596	6 (4.4)	7 (10.0)	0.122
Metabolic syndrome *	43 (34.1)	32 (53.3)	0.013	42 (31.1)	36 (51.4)	0.004

* Frequency (%).

**Table 4 metabolites-15-00237-t004:** The validation results for the serum uric acid cutoff point of >6.8 mg/dL indicate its efficacy in detecting each cardiometabolic factor and metabolic syndrome.

Cardiometabolic Factor	Sensitivity, %	Specificity, %	Correct Classification, %	Area Under the ROC Curve
Hyperglycemia	41.7	78.1	70.7	0.6292
Hypertriglyceridemia	36.4	82.5	62.0	0.6032
Altered HDL	26.2	75.3	36.4	0.5012
Insulin resistance	24.4	71.7	42.1	0.5001
Hypertension	35.0	74.5	72.5	0.5947
Metabolic syndrome	38.8	82.2	65.3	0.6601

## Data Availability

The data presented in this study are available upon request from the corresponding author due to privacy concerns.

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
