# Peer review of "Association of Serum Uric Acid Levels with Cardiometabolic Factors in Adolescents with Obesity: A Cross-Sectional Study"

_metabolites, 2025, doi:10.3390/metabo15040237_

Round 1
Reviewer 1 Report
Comments and Suggestions for Authors
The work is well written and may be considered for publication after revision. There are a few minor errors to re-consideration, namely:
1. Please correct the sentence: Uric acid is the final metabolic byproduct derived from both endogenous and exogenous purines.
2. Sex also influences SUA levels, although 48
a consensus for defining normal ranges has not been reached, with variations across different studies [7,8]. - please explain why, if available.
3. Mets or MetS?
4. line 72 - please provide names of mentioned pediatric centers.
5. BMI > 95th percentile based on the 2000 CDC growth charts the reference to appropriate guidelines is necessary.
6. Please rephrase: The odds ratios (ORs) for cardiometabolic factors and MetS among those in SUA T3 were estimated using 95% confidence intervals (CIs) via the Cochran–Mantel–Haenszel test.
7. 3.1. Generał Characteristics - the flow chart/graph of may be beneficial for manuscript and study understanding.
8. Discussion and conclusion sections are appropriate.
Author Response
1. Please correct the sentence: Uric acid is the final metabolic byproduct derived from both endogenous and exogenous purines.
R= We appreciate the comment, and the sentence has been corrected (line 50).
2. Sex also influences SUA levels, although a consensus for defining normal ranges has not been reached, with variations across different studies [7,8]. - please explain why, if available.
R= We appreciate the comment, and the likely justification is specified in the introduction: This sex difference can be explained by the fact that estrogens increase the renal clearance of urate and reduce the post-secretory tubular reabsorption of urate (line 56).
3. Mets or MetS?
R= We appreciate the comment, and the abbreviation has been standardized to MetS.
4. line 72 - please provide names of mentioned pediatric centers.
R= We appreciate the comment, and the names of the hospitals have been added. Hospital Infantil de Mexico Federico Gómez, Pediatric Hospital Centro Médico Nacional Siglo XXI, Pediatric Hospital of the National Medical Center West Guadalajara and High Specialty South Central Hospital of Petroleos Mexicanos (line 81).
5. BMI > 95th percentile based on the 2000 CDC growth charts the reference to appropriate guidelines is necessary.
R= We appreciate the comment, and reference #18 has been added (line 85)
6. Please rephrase: The odds ratios (ORs) for cardiometabolic factors and MetS among those in SUA T3 were estimated using 95% confidence intervals (CIs) via the Cochran–Mantel–Haenszel test.
R= We appreciate the comment, and the sentence has been corrected (line 140).
7. Generał Characteristics - the flow chart/graph of may be beneficial for manuscript and study understanding.
R= We appreciate the comment, and the Flow Chart has been added (line 158 & 210).

Reviewer 2 Report
Comments and Suggestions for Authors
In this manuscript (metabolites-3392599), the authors investigated the importance of serum uric acid levels (SUA) in predicting cardiometabolic risks in adolescents having obesity. The authors found that SUA levels are significantly correlated with specific cardiometabolic risks in adolescents with obesity. The authors have written the manuscript very well and understandably.
Major concerns
- For checking hyperglycemia, HbA1c is the best method.
- I was wondering about glucose levels. Hyperglycemia means above range 250 mg/dl for adolescents. I could see only normal levels here. Please justify.
- High uric acid levels are also an indicator of kidney problems. So, have authors ever noticed any markers or symptoms for renal dysfunction in these populations?
- Were additional cardiovascular disease markers, such as troponin, examined by the authors?
- The authors could have kept some number of normal adolescents to compare with these populations.
Language is considered good and understandable.
Author Response
1. For checking hyperglycemia, HbA1c is the best method.
R= We agree, and it will be added to the limitations that only fasting glucose was considered, and HbA1c was not included (line 273).
2. I was wondering about glucose levels. Hyperglycemia means above range 250 mg/dl for adolescents. I could see only normal levels here. Please justify.
R= According to the recommendations of the International Diabetes Federation, fasting glucose levels are considered abnormal when they exceed 100 mg/dL, and this cutoff point was used in our study. Regarding why most adolescents show normal glucose levels, patients receiving insulin sensitizers or insulin were excluded, which indirectly led to the exclusion of those with treated type 2 diabetes mellitus. Among the included participants, the 99th percentile for fasting glucose was 124 mg/dL.
3. High uric acid levels are also an indicator of kidney problems. So, have authors ever noticed any markers or symptoms for renal dysfunction in these populations?
R= We appreciate the comment. In general, it is well-known that it takes several years for renal alterations (such as lithiasis) or gout to develop in cases of hyperuricemia. Therefore, unless there is a genetic problem, such conditions are very rare in pediatric populations. Additionally, it is worth mentioning that all patients included in this study underwent evaluation of serum creatinine levels and a general urine test, and those with elevated creatinine levels or proteinuria were excluded. This information has been added to the methodology section (line 86).
4. Were additional cardiovascular disease markers, such as troponin, examined by the authors?
R= We appreciate your comment; however, other cardiovascular markers were not analyzed, and we believe this could be addressed in future studies.
5. The authors could have kept some number of normal adolescents to compare with these populations.
R= We appreciate your comment, but subjects without obesity were not analyzed. Therefore, it is mentioned in the limitations that these findings apply specifically to the population with obesity. “Another limitation is that by including only obese patients, the SUA cutoff point is specific but lacks sensitivity for detecting cardiometabolic factors, as observed in the results. Therefore, this cutoff can be applied only to adolescents with obesity” (Line 275).

Reviewer 3 Report
Comments and Suggestions for Authors
Anthropometry and Cardiometabolic profile assessment and uric acid level measurements needs more details
Author Response
Anthropometry and Cardiometabolic profile assessment and uric acid level measurements needs more details
R= We appreciate the comment, and the **Anthropometry and Cardiometabolic Profile** section has been further detailed (line 96)
